# News is More than a Collection of Facts:
# Moral Frame Preserving News Summarization

**Enrico Liscio†, Michela Lorandi‡ & Pradeep K. Murukannaiah†**
†Department of Intelligent Systems, Delft University of Technology
‡ADAPT Research Centre, Dublin City University
{e.liscio,p.k.murukannaiah}@tudelft.nl, michela.lorandi@adaptcentre.ie

## Abstract

News articles are more than collections of facts; they reflect journalists' framing, shaping how events are presented to the audience. One key aspect of framing is the choice to write in (or quote verbatim) morally charged language as opposed to using neutral terms. This moral framing carries implicit judgments that automated news summarizers should recognize and preserve to maintain the original intent of the writer. In this work, we perform the first study on the preservation of moral framing in AI-generated news summaries. We propose an approach that leverages the intuition that journalists intentionally use or report specific moral-laden words, which should be retained in summaries. Through automated, crowd-sourced, and expert evaluations, we demonstrate that our approach enhances the preservation of moral framing while maintaining overall summary quality.

## 1 Introduction

With shrinking attention spans in the social media era, fewer people read full news articles, favoring concise information, instead (Baqir et al., 2024). News summarization addresses this shift by condensing content into digestible formats. AI-driven summarization has long been studied (Zhang et al., 2025), with algorithms traditionally evaluated on fluency and coherence of the generated text, and faithfulness to the facts presented in the article (Fabbri et al., 2021). However, journalism researchers recognize that news articles are more than just a collection of facts—they contain intentional *framing* (De Vreese, 2005).

Framing refers to the way information is presented and reported to shape audience perceptions of an issue, e.g., by emphasizing certain aspects of an event while downplaying others (Entman, 1993). It explains how different news articles may present the same event in distinct ways. In this work, we focus on the *moral* dimension of framing, which evaluates actions, behaviors, or situations as right or wrong based on the underlying moral principles (Graham et al., 2013). When summarizing a news article, it is crucial to preserve not only the factual content but also the moral framing used to present it. Take the example in Figure 1, where the journalist chose to quote NGOs verbatim in their criticism of the US climate change plan. A neutral summary might state, *"The NGOs criticized the US plan"*, which, while factually accurate, overlooks the journalist's deliberate choice to convey the NGOs' strong condemnation. Rewording this in a neutral tone risks losing the original moral intent.

We perform the first exploration into the preservation of moral framing in AI-generated summaries. Precisely, we aim to identify a prompting strategy that best preserves moral framing but maintains overall summary quality. We leverage the zero-shot summarization ability of Large Language Models, shown to produce results on par with human-generated summaries (Goyal et al., 2022; Zhang et al., 2024). We compare three language models and five prompting methods. Leveraging the intuition that journalists intentionally use or report moral-laden words in the article text, we propose approaches that first identify moral-laden words in the article (e.g., through Chain-of-Thought or supervised classification) and then guide the language model in preserving such words in the summary.

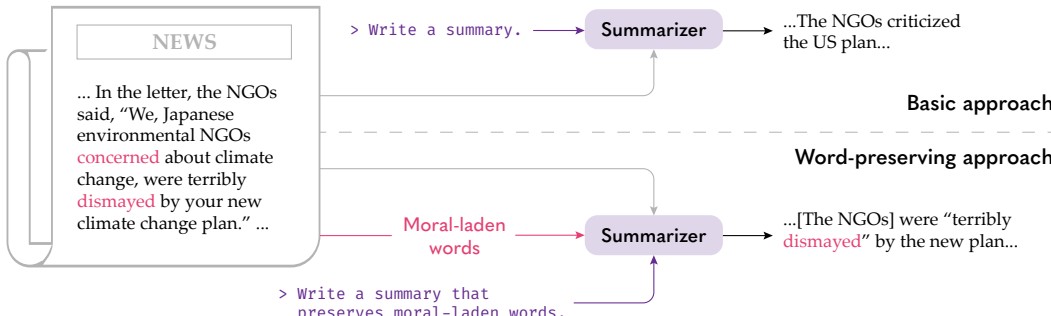

Figure 1: Example of moral framing, sourced from a 2002 Kyodo News article in the EMONA dataset (Lei et al., 2024). At the top, an example of how an AI-generated summary might overlook the moral framing in this segment. At the bottom, an example of how our proposed approach identifies the moral-laden words and preserves (part of) them in the summary.

We test our methods on the EMONA dataset (Lei et al., 2024), a collection of 400 news articles spanning different topics and political biases. We perform an extensive evaluation, including automated, crowd, and expert evaluations, to compare moral frame preservation across prompting methods and evaluate whether preserving moral framing affects overall summary quality. Our results indicate that maintaining moral-laden words in the summaries enhances moral framing preservation without reducing summary quality. The human evaluations show that moral frame preservation goes beyond automated evaluation metrics and requires human judgment, exposing unexplored factors (e.g., preserving quoted text in the summary) and risks (e.g., adding moral framing to neutral article segments).

## 2 Related works

We build on automated text summarization and moral framing in media, of which we provide an overview. We also review works on morality in the NLP field.

### 2.1 Automated text summarization

AI-based summarization has a long history (Zhang et al., 2025), initially relying on extractive summarization techniques that identify the most significant spans of text to include in the summary (Erkan & Radev, 2004; Mihalcea & Tarau, 2004). The development of deep learning and transformers shifted the focus to abstractive summarization, which generates summaries from scratch by fine-tuning pre-trained models, e.g., BERT (Liu & Lapata, 2019) or BART (Liu et al., 2022), on reference summaries. The advent of Large Language Models (LLMs) shifted the attention away from the fine-tuning paradigm. Large benchmarking studies on news summarization have shown that humans judge summaries generated by LLMs through zero-shot prompting to be comparable to human-written ones, and overwhelmingly prefer them over the reference summaries traditionally used to fine-tune language models (Goyal et al., 2022; Pu et al., 2023; Zhang et al., 2024). Recent work focuses on refining the summarization prompts, e.g., through Chain-of-Thought approaches that first prompt the model to list important facts and then integrate these facts into a summary (Wang et al., 2023) or that guide the model in retrieving relevant context (Liu et al., 2024).

### 2.2 Moral framing in media

Framing involves highlighting specific aspects of a news item to shape problem definitions, causal interpretations, moral judgments, and potential solutions (Entman, 1993). Different media outlets apply framing strategies to align narratives with ideological stances (Scheufele, 1999), appeal to specific audiences (Druckman, 2001), or reinforce societal norms (Jun et al., 2022). Framing and its effects have been widely studied across media and communication fields (Tversky & Kahneman, 1981; De Vreese, 2005). Computational approaches have

focused on moral framing in media, often through the lens of the Moral Foundation Theory (MFT) (Fulgoni et al., 2016; Mokhberian et al., 2020; Reiter-Haas et al., 2021), which postulates that human morality can be decomposed into five innate moral foundations such as fairness and loyalty (Graham et al., 2013). These studies have contributed to understanding political affiliations (Roy et al., 2021), media bias (Hamborg, 2023), vaccine hesitancy (Alonso del Barrio & Gatica-Perez, 2023), and violence against women (Mittal et al., 2024).

### 2.3 Morality in the NLP field

Modeling morality in text is gaining increasing attention in the NLP field (Reinig et al., 2024). Most works have focused on the detection of moral-laden text, by leveraging lexicons (Araque et al., 2020; Hopp et al., 2020), knowledge-graphs (Hulpuș et al., 2020; Asprino et al., 2022), or supervised machine learning (Liscio et al., 2022; Alshomary et al., 2022). The MFT has often been employed to model morality in NLP applications (Vida et al., 2023). Several datasets have been collected where the MFT foundations have been annotated on tweets (Hoover et al., 2020), Reddit posts (Trager et al., 2022), or prompt-reply pairs (Ziems et al., 2022). Several works have employed these datasets for a variety of applications, including domain adaptation (Huang et al., 2022), explainability (Liscio et al., 2023), sentence embeddings alignment (Park et al., 2024), and language generation (Dognin et al., 2025). However, such datasets contain annotations of text spans (e.g., labels on the full post). To our knowledge, the EMONA dataset (Lei et al., 2024), which we describe in detail in Section 4.1, is the only one containing news articles with word-level morality annotations.

## 3 Prompting Methods

To obtain an AI-generated summary, we provide a language model (referred to as *summarizer*) with the article text and prompt it to summarize it. We compare the five prompting methods in Table 1. All methods share the same basic structure, prompting the summarizer to generate a summary with a word limit of 200 (which we empirically identified as a trade-off between conciseness and enough space to preserve the article's moral framing).

| Category | Method | Approach |
|---|---|---|
| Basic prompts | *Plain* | The summarizer is prompted to write a summary of the article. |
| | *Direct* | The summarizer is prompted to write a summary that "preserves the moral framing of the original article". |
| Word-preserving prompts | *CoT* | The summarizer is first prompted to identify the list of moral-laden words and then to generate a summary that preserves them, à la Chain-of-Thought. |
| | *Oracle* | The summarizer is provided with the list of words annotated by humans as moral-laden in the article and asked to preserve them in the summary. |
| | *Class* | A classifier is trained on human annotations to identify moral-laden words in a news article. The summarizer is then prompted to generate a summary that preserves the words identified by the classifier. |

Table 1: An overview of the five prompting methods. The exact prompts are in Appendix A.

The basic methods simply prompt the summarizer to generate a summary, with *Direct* being a simple extension aimed at exploiting the summarizer's knowledge of moral framing. In contrast, the word-preserving methods leverage the intuition that journalists intentionally use or report moral-laden words in the article text, and maintaining them in the summaries helps preserve the intended moral framing. These methods extend *Direct* by providing the summarizer with a list of moral-laden words and requesting it to generate a summary that attempts to preserve as many of these words as possible (since not all words can be preserved due to the shorter nature of the summary compared to the article). The three approaches differ in how the list of moral-laden words is identified and in the resources required to generate the summary. *CoT* leverages the intuition of preserving moral-laden words without requiring additional resources. *Oracle* requires human annotations, and

is included as a practically infeasible but interesting topline. *Class* is a practical middle-ground option between *CoT* and *Oracle* that evaluates the importance and the efficacy of incorporating human annotations into an automated summarization pipeline.

# 4 Experiments

We describe our experimental procedure, starting with the dataset and models used[1].

## 4.1 Dataset

We experiment with the EMONA dataset (Lei et al., 2024), a collection of 400 news articles spanning different topics and political biases. In each sentence of the articles, all *event* instances were annotated by humans—where an event refers to an occurrence or action, considered the basic element in storytelling (Zhang et al., 2021)—yielding a total of over 10k sentences and 45k events. Each event was then labeled as moral-laden or neutral, with 21% (9.6k) of the events labeled as moral-laden. When labeled as moral-laden, the event was then annotated with one of the ten elements of the Moral Foundation Theory (such as *loyalty* or *fairness*). Examples and additional details are provided in Appendix D.3.1.

## 4.2 Models

As summarizers, we compare three state-of-the-art models that demonstrate strong general knowledge understanding and are instruction-tuned, as recommended by Zhang et al. (2024)—specifically, Llama-3-70B, Command R+, and DeepSeek R1.

For the *Class* method, we train Llama-3-8B to identify moral-laden words in the article text in a supervised fashion using the EMONA dataset (we employ a smaller model than the summarization models to keep training and validation manageable). We feed the article's sentences individually to the classifier which predicts moral-ladenness at a token level. We also tested two other approaches, namely (1) using the full article as input, and (2) a sequential approach that first identifies the sentences that contain moral-laden words and then the moral-laden words in them, but found them to perform worse. Hyperparameters and additional details are reported in Appendix B.

## 4.3 Summaries generation procedure

We split the dataset into training and test sets, which we treat as fixed sets to facilitate human evaluation. We select 60 news articles (15% of the EMONA dataset) as the test set, performing stratified sampling based on article source, topic, and article-level ideology stance, while maintaining a feasible size for the human evaluation.

To obtain the classifier for the *Class* method, we (1) perform a 3-fold cross-validation dividing the training set further into training and validation sets and choose the hyperparameters that lead to the best performance on the validation set, (2) train the model on the whole training set, and (3) perform inference on the test set's articles. We then generate summaries of the test set articles with the five methods and three summarization models. To account for variability, we repeat the generation five times with fixed seeds and report the average results. We randomly choose one of the five seeds and perform human evaluation on the summaries generated with that seed. For the word-preserving approaches, we treat all annotated moral-laden events as human-annotated moral-laden words, and all other article words as neutral. We do not distinguish among the ten MFT elements in the summary generation, but we do in the evaluation (by measuring moral divergence, see Section 4.4.1).

## 4.4 Evaluation

We perform automated, crowd, and expert evaluations of the summaries. Our evaluations are not intended to judge the overall quality of LLM-generated summaries, which has

---

[1]Code and results are available at: https://github.com/enricoliscio/moral-summarization

already been extensively studied (Goyal et al., 2022; Pu et al., 2023; Zhang et al., 2024). Instead, we employ these evaluations to compare moral framing preservation across methods and models, and assess whether instructing the summarizer to preserve moral-laden words compromises the overall quality of the summary. First, we employ five automated evaluation metrics to compare the generated summaries across summarization models. We use these results to choose the best-performing summarization model. Then, we perform crowd and expert evaluations of the summaries generated by the chosen model on the test set to investigate the aspects of moral framing preservation that are not captured by the automated metrics. Appendix C provides additional details on the human evaluations.

### 4.4.1 Automated evaluation

In line with a recent benchmarking survey (Zhang et al., 2024), we employ the following reference-free automated evaluation metrics to measure the quality of the summaries: (1) **SummaC** (Laban et al., 2022), which measures how the summary sentences are entailed from the news article; (2) **QAFactEval** (Fabbri et al., 2022), which measures entailment based on questions generated from the article and answers based on the summary; and (3) **BLANC** (Vasilyev et al., 2020), which measures the performance boost gained by a language model with access to the summary while performing a language understanding task on the article.

In addition, we introduce the following metrics to measure the preservation of the moral-laden words in the generated summaries: (1) *MoralCount*, which reports the number of words that were annotated as moral-laden in the article and that are preserved in the summary (to avoid double-counting, we employ the lemmas of the annotated and preserved words). (2) *MoralDivergence*, which measures the divergence between the distribution of moral annotations in the article and the summary. We normalize the distribution of moral annotations across the ten MFT moral categories and employ the Jensen-Shannon divergence to calculate the divergence between the distribution in the article and the summary (smaller divergence indicates more similarity between the distributions).

### 4.4.2 Crowd evaluation

We recruited crowd workers on the Prolific (www.prolific.co) crowdsourcing platform to evaluate the summaries generated by the best-performing model for the test set articles. The Ethics Committee of the leading author's university approved this study and we received informed consent from each subject. We ask each worker to evaluate two articles (and their summaries). To account for subjectivity and noise, we ensure that each article and its summaries are evaluated by two different workers and report the average.

We start by instructing the crowd worker on the concept of moral framing. Next, we show them a news article and ask them to highlight all spans of text that they deem morally framed. The decision to let the worker highlight spans of text instead of restricting to keywords is motivated by (1) avoiding repeating the annotation procedure performed in the EMONA dataset, and (2) allowing more freedom to the worker in line with typical frame analysis procedures in media studies (Matthes & Kohring, 2008). Next, we present the summaries generated via the five proposed methods one at a time (in random order to avoid the order effect). For each summary, we ask the worker to judge the extent to which each of the previously highlighted spans is preserved in the summary, on a 1–5 Likert scale.

### 4.4.3 Expert evaluation

We ask eight experts in the media and communication fields to provide quantitative and qualitative assessments of the summaries. Each expert is asked to evaluate three articles and the respective five summaries, resulting in 24 evaluated articles (out of the 60 that compose the test set). For each article, the expert is provided with all summaries (one summary per file, with the files randomly named to avoid biases) to allow comparisons. We ask the expert to rate each summary on a 1–5 Likert scale based on how well the summary preserves the moral framing of the original article. Then, we ask the expert to textually motivate the differences between summaries to which they have given different Likert scores.

| Metric | Model | *Plain* | *Direct* | *CoT* | *Oracle* | *Class* |
|---|---|---|---|---|---|---|
| QAFactEval (↑) | Llama-3-70B | 3.76 | 3.45 | 3.77 | 3.72 | 3.70 |
| | Command R+ | 3.55 | 3.08 | 3.38 | 3.40 | 3.40 |
| | DeepSeek R1 | 3.40 | 3.10 | 3.16 | 3.32 | 3.29 |
| SummaC (↑) | Llama-3-70B | 0.39 | 0.35 | 0.38 | 0.37 | 0.37 |
| | Command R+ | 0.36 | 0.32 | 0.35 | 0.34 | 0.34 |
| | DeepSeek R1 | 0.34 | 0.32 | 0.33 | 0.33 | 0.33 |
| BLANC (↑) | Llama-3-70B | 0.15 | 0.14 | 0.16 | 0.16 | 0.16 |
| | Command R+ | 0.15 | 0.14 | 0.16 | 0.16 | 0.16 |
| | DeepSeek R1 | 0.13 | 0.13 | 0.13 | 0.15 | 0.14 |
| *MoralCount* (↑) | Llama-3-70B | 5.85 | 6.49 | 7.31 | 10.02 | 8.75 |
| | Command R+ | 5.92 | 6.33 | 7.99 | 9.44 | 8.49 |
| | DeepSeek R1 | 4.99 | 5.65 | 6.37 | 10.18 | 8.20 |
| *MoralDivergence* (↓) | Llama-3-70B | 0.26 | 0.24 | 0.23 | 0.17 | 0.21 |
| | Command R+ | 0.27 | 0.26 | 0.22 | 0.19 | 0.21 |
| | DeepSeek R1 | 0.29 | 0.28 | 0.26 | 0.18 | 0.22 |
| Summary Length | Llama-3-70B | 153.0 | 161.4 | 152.4 | 174.6 | 173.9 |
| | Command R+ | 173.8 | 191.7 | 201.3 | 197.1 | 194.7 |
| | DeepSeek R1 | 146.8 | 162.2 | 159.2 | 174.3 | 173.2 |

Table 2: Results of the automated evaluation of the summaries of the test set articles. As detailed in Section 4.4.1, we report three reference-free automated evaluation metrics, two novel metrics to measure the preservation of moral-laden words, and the average length of the summaries. ↑ indicates that higher scores are better, ↓ indicates the opposite.

# 5 Results

We report the results of the automated, crowd, and expert evaluations.

## 5.1 Automated evaluation

Table 2 shows the automated evaluation results. First, we observe that there is a similar trend across models and methods with reference-free metrics (QAFactEval, SummaC, and BLANC). For all metrics, the results are comparable except for the *Direct* method, which yields slightly worse summaries. Thus, instructing the summarizer to preserve the identified moral-laden words does not affect the overall quality of the summary. When comparing the models, we observe that Llama-3-70B generally yields the best results, followed, in order, by Command R+ and DeepSeek R1. Based on these results, we decided to use Llama-3-70B to generate the summaries to be evaluated with the crowd and expert evaluations.

Second, on *MoralCount* and *MoralDivergence*, the methods rank consistently across models in the order: *Plain*, *Direct*, *CoT*, *Class*, *Oracle*. Thus, instructing the summarizer to preserve moral framing even without providing a list of moral-laden words (*Direct*) leads to better preservation of moral-laden words than just instructing to generate a summary (*Plain*).

Third, as expected, the word-preserving methods outperform the basic methods on *MoralCount* and *Moral-Divergence*. To compare the three methods, we first report the word-level moral-ladenness prediction performance. Recall (Table 1) that, in *CoT*, the summarizer was prompted to identify the moral-laden words to be preserved in the summary; in *Class*, a classifier was trained to detect moral-laden words in the article. The efficacy of these components is instrumental in the subsequent summary generation with the two methods. Table 3 reports their performance, measured by comparing the lists of predicted words and the list of words annotated as moral-laden in the

| | Model | $F_1$-score |
|---|---|---|
| *CoT* | Llama-3-70B | 22.3 |
| | Command R+ | 23.9 |
| | DeepSeek R1 | 22.8 |
| *Class* | Llama-3-8B | 47.2 |

Table 3: Performance of word-level moral-ladeness prediction with the *CoT* and *Class* methods.

EMONA dataset (thus, random prediction would lead to a near-zero $F_1$-score). We observe that *Class* leads to better moral-ladenness prediction results than *CoT*, which is reflected in the *MoralCount* and *MoralDivergence* results in Table 2. Unsurprisingly, *Oracle*—where the summarizer is provided with the list of words annotated as moral-laden—outperforms the other two word-preserving methods.

Finally, we observe that the requested maximum summary length (200 words) is generally respected but with variations across methods and models (with the average article length being 738.0 words). Command R+ generates consistently longer summaries than Llama-3-70B and DeepSeek R1, in several cases exceeding the word limit. We observe similar trends for Llama-3-70B and DeepSeek R1, with *Oracle* and *Class* leading to the longest summaries, likely due to the efforts in preserving the identified moral-laden words. However, despite the instructions to preserve moral-laden words, *CoT* results in shorter summaries than the other two word-preserving approaches. This is likely due to the number of moral-laden words that the summarizer is tasked to preserve in the summary—the median number of such words is 11 for *CoT*, 16.5 for *Class*, and 21.5 for *Oracle*. Not being tasked with these instructions, *Plain* and *Direct* lead to shorter summaries.

## 5.2 Crowd evaluation

Table 4 reports, for each method, the average Likert scores the workers gave to the morally framed spans they highlighted, and the differences between the scores assigned to the summaries generated by different methods.

|        | Plain | Direct | CoT | Oracle | Class | **Likert scores** |
|--------|-------|--------|------|--------|-------|-------------------|
| Plain  | -     | -      | -    | -      | -     | $3.15 \pm 0.66$   |
| Direct | 0.323 | -      | -    | -      | -     | $3.22 \pm 0.70$   |
| CoT    | 0.733 | 0.760  | -    | -      | -     | $3.20 \pm 0.74$   |
| Oracle | **0.030** | **0.036** | **0.037** | -   | -     | $3.33 \pm 0.64$   |
| Class  | **0.007** | **0.006** | **0.021** | 0.435 | -  | $3.40 \pm 0.67$   |

Table 4: Average per-article crowd Likert scores (shaded) and Wilcoxon pairwise comparisons between the methods, in bold the significantly different ones ($p < 0.05$) (unshaded).

First, we observe that *Class* and *Oracle* emerge as the best-performing methods with significant differences from the other three methods (as shown by the Wilcoxon pairwise comparison matrix in Table 4). However, we notice no significant difference between the two, although *Oracle* is more effective in preserving the annotated moral-laden words than *Class*, which in turn is more effective than *CoT* (as shown in Section 5.1). These results suggest that, when a sufficient moral-ladenness prediction performance is met, preserving moral-laden words in the summary improves the preservation of moral framing, but a better moral-ladenness prediction performance does not necessarily correlate with better moral framing preservation.

Second, we observe that all average scores range between 3 and 3.5 (rightmost column of Table 4). This is not surprising since these scores evaluate the extent to which morally framed article spans are preserved in the summary, and some are inevitably lost due to the summary's shorter length.

Third, we find no significant correlations between automated and crowd evaluation results (as reported in Table 5 together with the correlation to the summaries length). Thus, judging moral framing preservation requires human evaluation, as it (1) is not correlated to the summary quality, and (2) cannot be assessed by measuring the preservation of moral words.

Finally, we note that traditional inter-annotator agreement metrics are not applicable in our setting. For each article, annotators highlighted morally framed spans and rated each one on a Likert scale; these spans often differed significantly between annotators in both number and content. In addition, crucially, aligning on span selection was not the goal; highlighting served as a mnemonic tool to recall morally framed content. Our focus is on how well

|  | Plain | Direct | CoT | Oracle | Class |
|---|---|---|---|---|---|
| QAFactEval | 0.05 | 0.28 | 0.22 | 0.02 | 0.17 |
| SummaC | -0.08 | -0.15 | 0.15 | 0.06 | 0.28 |
| BLANC | 0.23 | 0.25 | 0.27 | 0.22 | 0.16 |
| *MoralCount* | -0.19 | -0.02 | -0.18 | -0.11 | 0.03 |
| *MoralDivergence* | -0.22 | -0.2 | -0.12 | 0.06 | -0.07 |
| Summary Length | 0.05 | 0.01 | -0.12 | 0.02 | 0.05 |

Table 5: Spearman correlation between per-article crowd and automated evaluations scores.

|  | Likert scores | # highlights |
|---|---|---|
| *Plain* | $3.63 \pm 1.08$ | $1.97 \pm 1.65$ |
| *Direct* | $3.54 \pm 1.03$ | $2.19 \pm 1.83$ |
| *CoT* | $3.66 \pm 1.01$ | $2.31 \pm 1.85$ |
| *Oracle* | $3.76 \pm 0.84$ | $2.74 \pm 2.15$ |
| *Class* | $3.71 \pm 1.01$ | $2.44 \pm 1.89$ |

Table 6: Crowd scores assigned to the highlights with at least one moral-laden word that is preserved in the summary (left) and the number of those spans (right).

annotators believe these moral frames are preserved in the summaries, as reflected in their Likert ratings. We explore a possible way to assess rating consistency in Appendix D.2.

**Preserving moral-laden words**  To evaluate the effect of preserving moral-laden words in the summary, we examine the scores given to the highlighted spans containing at least one annotated moral-laden word that is preserved in the summary. The spans containing moral-laden words are the same for all summaries—however, some summaries preserve these words while others do not. Thus, the number of these spans varies across methods. Table 6 reports the average scores assigned to such spans and their number. We observe that the scores are similar across methods and consistently higher than the average scores presented in Table 4, suggesting that maintaining moral-laden words promotes the preservation of moral framing. The difference across methods lies in the number of such highlighted spans, with *Class* and *Oracle* counting more of such spans (respectively, 24% and 39% more when compared to *Plain*), leading to the difference in the average scores observed in Table 4.

**Number of highlights per article**  The workers were allowed to highlight as many morally framed spans as necessary, resulting in a number of highlights ranging from 0 (in a single case) to 31, with an average of $6.6 \pm 4.7$. We correlate the number of spans a worker highlighted to the Likert scores they assigned to the summaries. We group the number of highlights per article in blocks of three and report in Figure 2 the average scores assigned by the workers to each method's summaries. The scores exhibit a downward trend—the more spans a worker highlighted, the lower the average scores they assigned to them. This is expected since the more text spans are highlighted, the more challenging it is for the summarizer to preserve them all in the summary. Next, we observe that *Class* generally outperforms the other methods across the numbers of highlights. In contrast, *Oracle* leads to the worst results in the 1-2-3 range but the best in the 4-5-6 range. We conjecture that this is due to the number of moral-laden words to be preserved in the summary (which is the highest for *Oracle*, as reported in Section 5.1)—attempting to preserve a large number of words may dilute the reporting of key aspects of the article, which is especially noticeable when only a few key morally framed text spans are highlighted and evaluated.

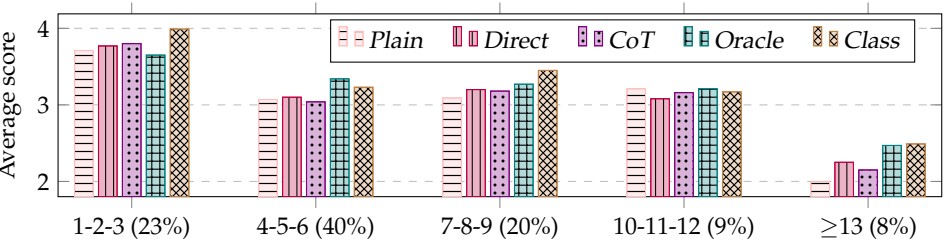

Figure 2: Average Likert scores grouped by the number of morally framed spans the workers highlighted in the article. The x-axis ticks indicate the number of highlighted spans (in parenthesis, the percentage of workers' assignments with that number of highlighted spans).

## 5.3 Expert evaluation

Table 7 reports the average expert Likert scores assigned to the summaries and the differences between the scores resulting from the different methods. We observe that *Plain* obtains significantly lower scores, but no significant difference among *Direct*, *CoT*, *Oracle*, and *Class*. This is likely due to the low number of samples (24) in the expert evaluation. Thus, we analyze the experts' motivations in support of their evaluations.

|  | *Plain* | *Direct* | *CoT* | *Oracle* | *Class* | **Likert scores** |
|---|---|---|---|---|---|---|
| *Plain* | - | - | - | - | - | $2.83 \pm 1.03$ |
| *Direct* | **0.004** | - | - | - | - | $3.46 \pm 0.96$ |
| *CoT* | **0.016** | 0.401 | - | - | - | $3.67 \pm 0.90$ |
| *Oracle* | **0.002** | 0.433 | 0.891 | - | - | $3.71 \pm 0.98$ |
| *Class* | **0.002** | 0.094 | 0.324 | 0.392 | - | $3.96 \pm 0.79$ |

Table 7: Average expert Likert scores (shaded) and Wilcoxon pairwise comparisons between different methods' scores, in bold the significantly different ones ($p < 0.05$) (unshaded).

The experts motivated the differences between each pair of summaries to which they gave different scores. We reviewed and categorized these motivations, assigning a label to each of the two summaries in the compared pairs. For instance, if an expert motivates the difference between the *Plain* and the *Class* summaries by writing that the former does not preserve a politician's moral-laden quote while the latter does, we assign a *Quote Omission* label to *Plain* and a *Quote Preservation* label to *Class*. Table 8 presents the categories we identified and their distribution across the different methods, of which we provide examples in Appendix D.3.3.

| Category | Label | *Plain* | *Direct* | *CoT* | *Oracle* | *Class* |
|---|---|---|---|---|---|---|
| **Positive** | Moral Framing Alignment | 23.7% | 20.4% | 29.8% | 13.9% | 68.3% |
|  | Quote Preservation | 2.6% | 8.2% | 2.1% | 11.1% | 17.1% |
|  | Examples Inclusion | 0.0% | 2.0% | 0.0% | 8.3% | 0.0% |
| **Negative** | Moral Framing Loss | 57.9% | 34.7% | 21.3% | 41.7% | 2.4% |
|  | Quote Omission | 2.6% | 8.2% | 12.8% | 5.6% | 0.0% |
|  | Examples Omission | 2.6% | 0.0% | 4.3% | 0.0% | 4.9% |
|  | Moral Framing Modification | 5.3% | 6.1% | 2.1% | 0.0% | 0.0% |
|  | Moral Framing Addition | 2.6% | 14.3% | 23.4% | 16.7% | 2.4% |
| **Neutral** | Similarity | 2.6% | 4.1% | 2.1% | 2.8% | 4.9% |

Table 8: Distribution of labels of the expert motivations of differences between summaries.

The *Class* method appears notably successful, with the highest proportion of positive labels (85.4%), particularly in moral framing alignment. In contrast, *Plain* receives the highest proportion of negative labels (57.9%), particularly in moral framing loss. Next, we discuss the two negative labels that do not have a positive counterpart, moral framing *modification* and *addition*. Modification indicates changes in how a morally framed article segment is presented, e.g., emphasizing different moral aspects. Addition occurs when a neutral article segment is presented with moral framing by incorporating moral-laden language not found in the article. These labels appear in over 20% of the evaluations of the *Direct* and *CoT* summaries, highlighting the risk that, without proper guidance, summarization models may inadvertently project their internal moral biases onto neutral segments.

**Preserving quotes** The categories above are challenging to quantify by other means, underscoring the importance of expert evaluation. However, the presence of quotes—that is, spans of text that report verbatim a spokesperson's words—can be quantified. In the test set articles, on average, 9% of the text is composed of quotes, but 16% of words annotated as moral-laden fall within quotes. Table 9 reports the occurrence of quotes in the text of the article and of the summaries, along with the subset of quotes that contain a word annotated as moral-laden. We observe that the word-preserving approaches contain significantly more quotes and quotes containing moral-laden words than the basic approaches.

|        | # quotes        | # w/ moral words |
|--------|-----------------|------------------|
| *Plain*  | $1.15 \pm 1.41$ | $0.38 \pm 0.78$ |
| *Direct* | $1.32 \pm 1.27$ | $0.55 \pm 0.92$ |
| *CoT*    | $3.98 \pm 3.15$ | $1.73 \pm 2.07$ |
| *Oracle* | $3.30 \pm 3.37$ | $1.82 \pm 2.57$ |
| *Class*  | $3.37 \pm 3.46$ | $1.40 \pm 2.10$ |
| Article | $4.27 \pm 4.39$ | $2.33 \pm 2.95$ |

Table 9: Average number of quoted spans present in the summaries and original article (left), and number of such spans that contain at least one moral-laden word (right).

|        | Likert scores   | # highlights    |
|--------|-----------------|-----------------|
| *Plain*  | $3.51 \pm 1.30$ | $0.40 \pm 0.73$ |
| *Direct* | $3.65 \pm 1.40$ | $0.42 \pm 0.76$ |
| *CoT*    | $3.44 \pm 1.49$ | $0.47 \pm 0.84$ |
| *Oracle* | $3.63 \pm 1.03$ | $0.55 \pm 0.94$ |
| *Class*  | $3.57 \pm 1.21$ | $0.53 \pm 0.91$ |

Table 10: Average crowd scores assigned to the highlights containing at least one span of quoted text containing at least one moral word that is preserved in the summary (left) and the number of those highlights (right).

Further, we observe that 20% of the spans highlighted by crowd workers contain at least one quote. Thus, we measure how the workers evaluated such spans. Table 10 reports (as a subset of the results in Table 6) the number of highlighted spans that contain at least one quote which, in turn, contains one word that was annotated as moral-laden, and the scores that workers assigned to such spans. We observe that, in line with the results in Table 6, the scores are higher than the average scores (Table 4) and that *Class* and *Oracle* lead to a larger number of such spans—respectively, 33% and 38% more when compared to *Plain*.

## 6   Conclusions

Moral framing is an integral component of news. We introduce and evaluate prompting strategies to guide Large Language Models in generating summaries that preserve moral framing by identifying and retaining moral-laden words from the article text. Through a crowdsourced evaluation, we show that humans perceive our approach as enhancing moral framing preservation while maintaining overall summary quality. Finally, an expert evaluation offered additional qualitative insights, highlighting the importance of preserving quoted text and the risks associated with adding or altering moral framing.

Explaining how the summarization models identify moral-laden words and decide which ones to preserve is instrumental to improving transparency to users, especially considering the inherent subjectivity of morality (van der Meer et al., 2024). Reducing reliance on human annotations can foster the approach's practicality, e.g., by enhancing the Chain-of-Thought method with the support of few-shot examples or moral lexicons, or by evaluating how LLM judges can approximate human judgements on this task. Finally, the EMONA dataset offers further avenues of exploration, e.g., our results could be correlated with the political bias of the article's source, and the prompting methods could be extended by integrating the Moral Foundation Theory annotations associated with moral-laden words.

## Acknowledgments

This material is produced as part of AlgoSoc, a collaborative 10-year research program on public values in the algorithmic society, and the Hybrid Intelligence Centre, both funded by the Dutch Ministry of Education, Culture and Science (OCW) under the Gravitation programme (project numbers 024.005.017 and 024.004.022). Any opinions, findings, and conclusions or recommendations expressed in this material are those of the author(s) and do not necessarily reflect the views of OCW or those of the AlgoSoc consortium as a whole. Research reported in this work was partially or completely facilitated by computational resources and support of the Delft AI Cluster (DAIC) at TU Delft (RRID: SCR_025091), but remains the sole responsibility of the authors, not the DAIC team. Michela Lorandi's work was conducted with the financial support of the Science Foundation Ireland Centre for Research Training in Digitally-Enhanced Reality (d-real) under Grant No 18/CRT/6224 and the ADAPT SFI Research Centre at Dublin City University, funded by the Science

Foundation Ireland under Grant Agreement No. 13/RC/2106_P2. A special thanks to Mohammed Al Owayyed for his support with setting up the crowd task.

## Ethics statement

Preserving the moral framing of news articles may inadvertently reinforce strong stereo­typical narratives, some of which could be disturbing or polarizing for readers. This raises concerns about the potential amplification of biases embedded within the original con­tent. Additionally, the identification and preservation of moral-laden content is inherently influenced by the moral compass of the underlying models, which are trained on data pre­dominantly sourced from WEIRD (Western, Educated, Industrialized, Rich, and Democratic) populations. As a result, certain moral perspectives may be overrepresented, while others remain underexplored. This can become especially apparent in cases of moral framing addition to the content of the original article, as unveiled by our expert evaluation. To over­come this, we envision integrating our approach with bias mitigation techniques (Gallegos et al., 2024)—by presenting both framing-sensitive and neutralized summaries, readers can gain a comprehensive understanding of diverse moral perspectives. Such a pairing can help increase awareness of the opinion landscape, encourage critical engagement with news content, and expose ideological diversity. Finally, our work remains an exploratory investigation conducted on an English-language dataset, primarily reflecting US-centric content. Further research is necessary to evaluate the robustness and applicability of our approach across different languages, cultural contexts, and minority perspectives.

## Reproducibility statement

We provide all the necessary details for reproducing our experiments throughout the paper and the Appendix. Methods, prompting, and training procedures are described in Sections 3 and 4. Appendix A reports the complete prompts used to generate the summaries. Appendix B provides additional details on our experiments, including the tested and used hyperparameters and computational infrastructure. Appendix C presents additional information on the human evaluations, including the aggregated demographics and further details on the crowd task. Code, generated summaries, and (anonymized) automated, crowd, and expert evaluation results are available on GitHub[1]. The model weights trained with the *Class* method are available in a public repository (Liscio et al., 2025a).

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

# A    Prompts

We report the complete version of the prompts used to generate summaries.

> **▍** *Plain*
>
> You have to summarize the following article.
> Here is the news article:
> [Article text]
> The summary has to be returned after a "SUMMARY:" token and ending with a "END OF SUMMARY." token. The summary should be no longer than 200 words.

> **▍** *Direct*
>
> You have to summarize the following text preserving the moral framing that the author gave to it.
> Here is the news article:
> [Article text]
> The summary has to be returned after a "SUMMARY:" token and ending with a "END OF SUMMARY." token. The summary should be no longer than 200 words.

> **▍** *CoT*
>
> You have to summarize the following text preserving the moral framing that the author gave to it.
> Here is the news article:
> [Article text]
> (1) First, you identify all the single words that are morally framed. Identify this step as "STEP 1:" and report each word in a new line starting with *
> (2) Finally, you write a summary of the news article. Please preserve as many morally framed words as possible in your summary. The summary has to be returned after a "SUMMARY:" token and ending with a "END OF SUMMARY." token. The summary should be no longer than 200 words.

> **▍** *Oracle*
>
> You have to summarize the following text preserving the moral framing that the author gave to it.
> Here is the news article:
> [Article text]
> The author used the following morally framed words in the article:
> [Bullet point (*) list of words annotated in the EMONA dataset]
> Please preserve as many morally framed words as possible in your summary. The summary has to be returned after a "SUMMARY:" token and ending with a "END OF SUMMARY." token. The summary should be no longer than 200 words.

> **▍** *Class*
>
> You have to summarize the following text preserving the moral framing that the author gave to it.
> Here is the news article:
> [Article text]
> The author used the following morally framed words in the article:
> [Bullet point (*) list of words identified by the supervised classifier]
> Please preserve as many morally framed words as possible in your summary. The summary has to be returned after a "SUMMARY:" token and ending with a "END OF SUMMARY." token. The summary should be no longer than 200 words.

### A.1 Alternative prompts tested

We experimented with several prompt variants, evaluating them heuristically and through the automated evaluation metrics. When observing similar results, we opted for the simpler solution, which is the one used in the paper experiments. The following are the tested variants (in parenthesis, the comparison with the prompts used in the paper experiments):

- Prompt the summarizer to generate a summary that is *"coherent, fluent, and only comprises relevant content that is consistent with the article"*, in line with the evaluation metrics proposed by Fabbri et al. (2021). (similar results)

- In the *CoT* method, prompt the summarizer to (1) extract the moral-laden words, (2) identify the sentences where the extracted words are, (3) generate the summary. (similar results)

- In the *Oracle* and *Class* methods:
  - instead of providing the moral-laden words as a list, highlight them throughout the article text (e.g., *"This is an example [moral-laden] of the text of an article [moral-laden]"*, where *"example"* and *"article"* were annotated/identified as moral-laden) (worse results)
  - Along with the moral-laden word, specify in which sentence the word was used in the article (*"\* In the nth sentence, the author used the moral-laden word [insert-word]"*) (worse results)
  - Provide the list of moral-laden words in brackets instead of bullet points (*"[word1, word2, word3, ...]"*) (similar results)
  - Use the lemmas instead of the annotated/identified words (similar results)

## B Experimental Details

We provide additional details on the models and the infrastructure used in our experiments.

### B.1 Models

We present the hyperparameters used for the summarizer models. Next, we describe the approaches we compared for moral-ladenness token-level classification for the *Class* method, together with the tested and used hyperparameters.

#### B.1.1 Summary generation

Table B1 lists the hyperparameters used for the summarization models. If a parameter is not present in the table, the default value supplied by the framework is used. The system prompt is set to *"You are a news summarizer assistant"* for the *Plain* method and to *"You are a news summarizer assistant and a moral expert"* for the other four methods.

| Model | Full name | Temperature | Top-p |
|-------|-----------|-------------|-------|
| **Llama-3-70B** | meta-llama/Meta-Llama-3-70B-Instruct | 0.6 | 0.9 |
| **Command R+** | CohereForAI/c4ai-command-r-plus-4bit | 0.6 | 0.9 |
| **DeepSeek R1** | deepseek-ai/DeepSeek-R1-Distill-Qwen-32B | 0.6 | 0.9 |

Table B1: Hyperparameters used for the zero-shot summarization.

#### B.1.2 Morality classification

We compare three approaches to token-level classification of moral-ladenness. (1) **Article** uses the full article as input and predicts moral-ladenness for each token; (2) **Sentence** feeds each article's sentences individually to the model, which predicts moral-ladenness at a token level; and (3) a **Sequential** approach where we first identify the sentences containing moral-laden words (sequence classification) and then identify the moral-laden words in

them (token classification). The latest was inspired by the fact that only 54% of the articles sentences contain at least one moral annotation.

We compare the three approaches by training Llama-3-8B (meta-llama/Llama-3-8B). The *Article* approach performed consistently worse than the others, likely due to the small number of articles in the dataset (400) compared to the number of sentences (over 10k). Thus, we decided to compare only the *Sentence* and the *Sequential* approaches through a 3-fold cross-validation, which we also use to select the best hyperparameters. Table B2 shows the hyperparameters that were compared in this setting, highlighting in bold the best-performing option and the resulting $F_1$-score. If a parameter is not present in the table, the default value supplied by the framework is used. The results show that directly performing moral-ladenness classification at the sentence level produces better results when compared to using a sequential approach. Thus, we use the *Sentence* approach in the paper.

| | *Sentence* | *Sequential* | |
| --- | --- | --- | --- |
| **Hyperparameters** | Token Classification | Sequence Classification | Token Classification |
| Epochs | **3**, 4 | 3, **4** | 3, **4** |
| Learning rate | **0.0002**, 0.0005 | **0.0002**, 0.0005 | 0.0002, **0.0005** |
| Learning rate scheduler | **linear**, cosine | linear, **cosine** | **linear**, cosine |
| Batch size | **8**, 16 | 8, **16** | 8, **16** |
| LoRA rank | 32, **64** | **32**, 64 | 32, **64** |
| LoRA dropout | **0.05** | **0.05** | **0.05** |
| LoRA $\alpha$ | **16** | **16** | **16** |
| Best $F_1$-score | $45.5 \pm 0.8$ | $78.5 \pm 0.5$ | $44.2 \pm 0.7$ |

Table B2: Hyperparameters tested and selected (in bold) when comparing the *Sentence* and *Sequential* token-level moral-ladenness classification approaches, with the resulting best $F_1$-score on the validation set.

## B.2 Computing and software infrastructure

The Llama-3-8B token-level moral-ladenness classifier was trained with the procedure described in Section 4.3 and B.1.2 on one NVIDIA A40 48GB GPU with 4-bit quantization. The hyperparameter tuning described in Table B2 took 60 hours. The summarization generation procedure described in Section 4.3 was conducted on one NVIDIA A100 80GB GPU with 4-bit quantization. In our experiments, we fixed five seeds (49, 311, 345, 655, 897) to ensure reproducibility. In total, the generation of all texts for all seeds took 21 hours for Llama-3-70B, 60.5 hours for Command R+, and 96 hours for DeepSeek R1. The libraries used in the experiments are listed as requirements in the code.

## C Human evaluations

We provide additional details on the crowd and expert evaluations. All exact instructions and informed consent forms are provided as supplemental material (Liscio et al., 2025b).

### C.1 Crowd evaluation

We provide additional information on the crowd evaluation described in Section 4.4.2.

**Annotation job layout** We hosted our evaluation task on the Qualtrics[2] platform. After obtaining informed consent, we introduced the workers to the concept of moral framing and provided a brief tutorial on the highlighting mechanism by instructing the workers to highlight spans of a mock article with their cursor. Next, they were shown the spans they had highlighted under a mock summary and instructed on the sliding mechanism, to be used to indicate the extent to which the corresponding highlighted morally framed span is

---

[2]www.qualtrics.com

preserved in the summary, ranging from 1 (Not Present) to 5 (Clearly Present). Subsequently, they were shown an article followed sequentially by its five summaries. They were asked to highlight all spans of article text they deemed morally framed. For each highlighted span, a slider would be created under each summary, which they would use to indicate the extent to which the corresponding highlighted span of text was preserved in that summary.

**Quality control**   The crowd workers were required to be fluent in English and have submitted at least 100 Prolific jobs with at least a 95% acceptance rate. Furthermore, due to the US-centric nature of several articles in the test set, we also required the workers to be residents of the US. We included four control tasks in each assignment. Two were in the tutorial, with the workers being asked to highlight exactly two spans of text and use the sliders to indicate that the first is clearly present in the mock summary (5 on the Likert scale) and the second not present (1 on the Likert scale). Next, under one of the summaries shown for each of the two articles, an extra slider was generated with a text requesting the workers to move it to the leftmost position (Not present) (for a total of two additional control tasks). As per Prolific guidelines, we allowed workers to fail at most one control task.

**Completion and payment**   78 workers completed the task, with 15 discarded for failing the control tasks. We planned to collect exactly two annotations per article. However, due to the distribution of the annotation tasks on the Qualtrics platform and workers accepting but returning the job before completing it, we collected six more annotations than required. We paid all workers and discarded the extra annotations. Finally, we obtained exactly two annotations per article, from a total of 62 workers. Each worker was paid £6.75 for an expected assignment duration of 45 minutes (at the rate of £9/h as per Prolific suggestion of fair retribution). Ultimately, the average time spent by a crowd worker on an assignment was $40.5 \pm 24.5$ minutes. Figure C1 reports the demographics of the workers whose submissions were included in the study (as self-reported in the Prolific platform).

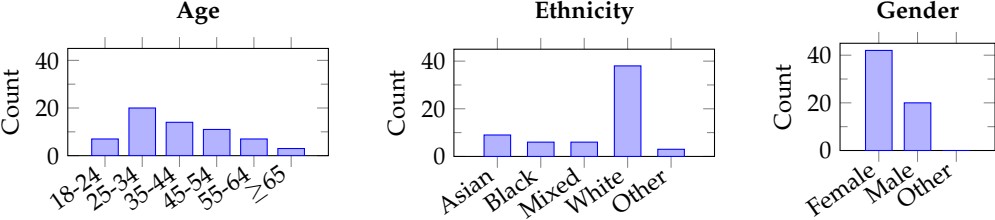

Figure C1: Demographics of crowd workers.

## C.2   Expert evaluation

The eight experts were contacted through the authors' network. They are all researchers (including four PhD, three postdoctoral researchers, and one assistant professor) in the age range 26-35 and based in Europe, conducting research in social sciences in the media domain, including behavioral sciences, political communication, and philosophy. We obtained informed consent from all experts.

# D   Extended results

We report additional automated evaluation results, and insightful examples from our summaries and the expert evaluation.

## D.1   Automated evaluation

We generate summaries with the *Plain*, *Direct*, *CoT*, and *Oracle* methods for all articles in the dataset, and with *Class* for the articles in the test set. In Section 5.1 we report the automated evaluation on the test set, while Table D1 shows the results on the full dataset

| Metric | Model | *Plain* | *Direct* | *CoT* | *Oracle* |
|---|---|---|---|---|---|
| QAFactEval (↑) | Llama-3-70B | 3.72 | 3.33 | 3.68 | 3.65 |
| | Command R+ | 3.42 | 2.99 | 3.28 | 3.33 |
| | DeepSeek R1 | 3.33 | 3.06 | 3.17 | 3.29 |
| SummaC (↑) | Llama-3-70B | 0.40 | 0.34 | 0.37 | 0.37 |
| | Command R+ | 0.35 | 0.31 | 0.34 | 0.34 |
| | DeepSeek R1 | 0.34 | 0.32 | 0.33 | 0.34 |
| BLANC (↑) | Llama-3-70B | 0.16 | 0.15 | 0.16 | 0.17 |
| | Command R+ | 0.15 | 0.14 | 0.16 | 0.16 |
| | DeepSeek R1 | 0.13 | 0.13 | 0.14 | 0.15 |
| *MoralCount* (↑) | Llama-3-70B | 5.75 | 6.20 | 7.06 | 9.86 |
| | Command R+ | 5.48 | 5.89 | 7.30 | 9.13 |
| | DeepSeek R1 | 4.86 | 5.22 | 6.24 | 9.86 |
| *MoralDivergence* (↓) | Llama-3-70B | 0.25 | 0.24 | 0.22 | 0.16 |
| | Command R+ | 0.26 | 0.25 | 0.22 | 0.18 |
| | DeepSeek R1 | 0.27 | 0.26 | 0.24 | 0.16 |
| Summary Length | Llama-3-70B | 153.9 | 163.0 | 151.5 | 175.3 |
| | Command R+ | 172.9 | 189.6 | 196.2 | 193.9 |
| | DeepSeek R1 | 146.9 | 159.1 | 159.5 | 172.9 |

Table D1: Results of the automated evaluation of the summaries of the complete dataset. As detailed in Section 4.4.1, we report three reference-free automated evaluation metrics, two novel metrics to measure the preservation of moral-laden words, and the average length of the summaries. ↑ indicates that higher scores are better, ↓ indicates the opposite.

(except for the *Class* method, for which the evaluation is only performed on the test set). The average *MoralCount* is 19.42 in the complete dataset and 19.08 in the test set. The average article length in the complete dataset is 719.73. We observe that the automated evaluation results' trends in the test set align with those in the full dataset. Thus, since the generation of summaries with the *Class* method and the human evaluation are performed only on the test set, in the paper we discuss the results on the test set.

## D.2   Inter-annotator agreement

As noted in Section 5.2, traditional inter-annotator agreement metrics are not appropriate in this context, as annotators highlighted and rated different text spans. Instead, we offer a proxy measure to approximate agreement. To illustrate this, we walk through a hypothetical annotation example that clarifies the process in detail. Table D2 shows a fictitious example of the crowd annotations for the summaries generated through one of the five methods.

| Art. ID | Ann. ID | Likert scores | Mean ann. score | Mean art. score | SD art. score |
|---|---|---|---|---|---|
| 0 | a | [2,2,3,5] | 3 | 3.17 | 0.17 |
| 0 | b | [3,3,4] | 3.33 | | |
| 1 | c | [3,4,5] | 4 | 3.6 | 0.4 |
| 1 | d | [1,3,5,4,3] | 3.2 | | |
| ⋮ | ⋮ | ⋮ | ⋮ | | |
| 59 | x | [2,5] | 3.5 | 2.95 | 0.55 |
| 59 | y | [3,3,3,1,2] | 2.4 | | |

Table D2: Fictitious example of the crowd annotations for the 60 test set summaries generated through one of the five prompting methods.

In this example, two annotators (a and b) annotated article 0. Each of the two annotators highlighted a different number of morally framed spans in the article (4 and 3, respectively)

and thus assigned different numbers of Likert scores. Calculating the inter-annotator agreement of these scores is not possible, as these scores were assigned to different items (i.e., different text spans). We then average these Likert scores, resulting in one score per annotator (3 and 3.33, respectively). Then, we average the two annotators' scores into one article score (3.17). We perform this for all 60 articles in the test set, and report in Table 4 the mean and standard deviation of these 60 scores (in this fictitious case, we would report the mean and standard deviation of the 60-dimensional vector [3.17, 3.6, ..., 2.95]).

Additionally, we can calculate the standard deviation between the two mean annotator scores assigned to one article (in the case of article 0 in this example, the standard deviation of the vector [3, 3.33], which is 0.17). We can then compute the mean and standard deviation of the standard deviation vector (in the fictitious example above, of the vector [0.17, 0.4,...,0.55]).

Table D3 reports the mean and standard deviation of the standard deviation vector of our experimental results across the five methods. These results show that the *Plain* approach leads to larger differences in average Likert scores among annotators when compared to the other approaches, which are comparable to each other. Furthermore, the magnitude of the mean standard deviation appears relatively low (although difficult to judge as no baseline is available), hinting at a consistency among annotators.

| Method | Mean SD |
|--------|---------|
| *Plain* | $0.80 \pm 0.53$ |
| *Direct* | $0.59 \pm 0.47$ |
| *CoT* | $0.59 \pm 0.46$ |
| *Oracle* | $0.55 \pm 0.45$ |
| *Class* | $0.57 \pm 0.48$ |

Table D3: Mean and SD of the SD vector of the results.

### D.3 Examples

We provide additional examples of annotations in the EMONA dataset, misclassification of moral-laden words, and motivations provided by the experts in support of their evaluations.

#### D.3.1 Annotation examples

Table D4 displays two additional examples of event annotations in the EMONA dataset, adapted from the original paper (Lei et al., 2024), similar to the example shown in Figure 1. Table D5 lists the ten MFT moral foundations used to annotate moral-laden events.

**Annotation examples**

In a succinct speech [non-moral], the new president Donald Trump told [non-moral] Americans: "The time for empty talk [cheating] is over. Now arrives the hour of action [authority]."

Texas Gov. Rick Perry is on the attack [non-moral], claiming [non-moral] in a TV ad that President Obama is waging a war [degradation] on religion, and he is the GOP candidate that can defend [purity] faith in America.

Table D4: Example annotations in the EMONA dataset. In brackets, non-moral, virtue, and vice annotations.

| Foundation | Definition |
|------------|------------|
| Care/ Harm | Support for care for others/ Refrain from harming others |
| Fairness/ Cheating | Support for fairness and equality/ Refrain from cheating or exploiting others |
| Loyalty/ Betrayal | Support for prioritizing one's inner circle/ Refrain from betraying the inner circle |
| Authority/ Subversion | Support for respecting authority and tradition/ Refrain from subverting authority or tradition |
| Purity/ Degradation | Support for the purity of sacred entities/ Refrain from corrupting such entities |

Table D5: The MFT moral foundations (virtue/vice).

#### D.3.2 Moral-laden word preservation in the summary

Figure 1 provides a successful example of moral-laden word preservation in the summary in our test set. However, we observe that not all quotes present in the summaries correspond to quotes in the article. For instance, in the same article from which the example in Figure 1 is sourced, the journalist reports another quote from the protesters, *"'We cannot accept it by any means. It is very regrettable that the United States has taken a negative stand [...]'"*. This passage is referred to by the *CoT* summary with *'Environmental groups staged a protest against*

*the U.S. alternative [. . . ], calling it a "regrettable" and "negative" stand'*, where *"regrettable"* and *"negative"* were mistakenly identified as moral-laden and reported in two separate quotes.

### D.3.3 Expert evaluation

The experts compared pairs of summaries, motivating the reasoning behind their evaluations. Table D6 reports examples of such motivations for different categories identified in our analysis (referring to the compared summaries as SUMMARY A and SUMMARY B). The most prevalent type of positive motivation is *Moral Framing Alignment*, where the summary's moral framing closely reflects that of the article. Positive motivations also emphasize the summary's fidelity to the article, particularly in direct *Quote Preservation* and *Example Inclusion*. Negative motivations primarily concern discrepancies in moral framing between the summaries and the article. *Moral Framing Loss* captures the cases where the article's moral framing is weakened or entirely omitted. Other negative evaluations emphasize the absence of key quotes (*Quote Omission*) or examples (*Examples Omission*). In *Moral Framing Modification*, summaries simultaneously omit and introduce different moral framing. Finally, *Moral Framing Addition* refers to cases where summaries introduce moral framing not present in the original article.

| Category | Label | Examples |
|---|---|---|
| **Positive** | Moral Framing Alignment | *SUMMARY A seems to present ACLU's position as if they were the ones in the article, [...]* 
 *SUMMARY A is more balanced and reflective of the original moral frame by incorporating defense from Newsweek: "Newsweek has defended the photo, saying it was one of several similar images taken of Bachmann." while [...]* |
| | Quote Preservation | *SUMMARY A preserves the quote from Walmart's governor "threatened to undermine the spirit of inclusion" [...]* 
 *SUMMARY A mentions "impose a Latin ban", "stupid and cruel", "huge moral failure."* |
| | Examples Inclusion | *SUMMARY A omits personal examples; SUMMARY B keeps them.* 
 *[...]; SUMMARY B keeps them.* |
| **Negative** | Moral Framing Loss | *SUMMARY A gives space only to the first moral framework (the judge is wrong), without adequately presenting the second one.* 
 *SUMMARY A loses the speculative tone of the text and presents everything as factual. [...]* |
| | Quote Omission | *SUMMARY A doesn't mention "twisted ideology" that promotes "unconscionable acts of violence and hate."* 
 *SUMMARY A doesn't contain the following extensive elaboration on the differences: "the move is seen as a significant escalation of presidential power on immigration policy. [...]"* |
| | Examples Omission | *SUMMARY A omits mention of the rarity of a former President's public criticism and examples, making it weaker.* 
 *SUMMARY A omits mention of the rarity of a former President's public criticism and examples; [...]* |
| | Moral Framing Modification | *[...] while SUMMARY B omits this part. Besides, SUMMARY B added too much moral-related phrasing such as "fair and respectful media representation of women in politics." and "conservative critics like Brent Bozell" which is not accurately reflected in the original article.* |
| | Moral Framing Addition | *[...] SUMMARY B is in the other extreme, it keeps the speculative framing, but adds that this represents "a blow for the Trump campaign", which is not said in the original article.* 
 *[...] SUMMARY B keeps some of the moral framing, but it does so by adding extra layer that change the overall meaning of the revelations."* |

Table D6: Examples of motivations the experts provided when evaluating the summaries.

