# OpenReview forum: "News is More than a Collection of Facts: Moral Frame Preserving News Summarization"
_colmweb.org/COLM/2025/Conference — COLM 2025_

### Official Review · Reviewer_iyXe · 2025-05-10

**Rating:** 7
**Confidence:** 4
**Ethics Flag:** 1

**Summary:**

The paper is the first study on preserving moral framing in LLM-generated summaries of news articles. It compares several prompting approaches and demonstrates that enhancing the prompt with automatically-identified moral-laden words leads to summaries that better preserve the article's moral framing. Evaluation is based on automatic metrics, crowdsourcing and expert annotation.

The work addresses a novel and interesting aspect of news summaries. making a valuable contribution to their evaluation. The proposed approach, which is based on identifying expressions that need to be preserved in the summary is somewhat straightforward, but reasonable and effective. The paper is clearly written and the experiments are thorough and well designed.

It would have been interesting to test a lexicon-based approach for identifying moral-laden words, as it would save the need for a fine-tuned classifier. It would also be good to know how well LLM judges are able to approximate human judgements on this task, to facilitate automatic evaluation of moral framing preservation.

Minor comments:

* It would be good to provide some data on inter-annotator agreement.

* Explain what is measured by each of the metrics in Table 2.

**Reasons To Accept:**

* The work explores a novel dimension of news summaries quality - moral framing preservation.

* It proposes a method that improves summaries in this respect.

* The paper is clearly written and the experiments are thorough and well designed.

**Reasons To Reject:**

* The paper could be improved by evaluating lexicon-based prompting and LLM-as-a-judge, as detailed in the summary above.

---

> ### Author Response · Authors · 2025-06-01
> **Rebuttal**
>
> We thank the reviewer for the interesting suggestions.
>
> - **It would have been interesting to test a lexicon-based approach for identifying moral-laden words, as it would save the need for a fine-tuned classifier.**
>
> This is an interesting suggestion. Since the aim of our work was to perform the first fundamental exploration into the preservation of moral framing in LLM-generated summaries, we opted for supervised classification as it is shown to lead to better performance in detecting moral-laden words, as reported in Section 2.3 and, e.g., in the results of the ValueEval Shared Task [1,2]. Nevertheless, **the use of a lexicon-based approach would fit nicely between *CoT* and *Class* in our scale of resource requirements** (lines 108-110), as it requires some resources (the lexicon) but no additional training. Employing lexicon-based approaches would, however, introduce an interesting complication: such approaches typically return a **continuous score of moral-ladenness** (e.g., Araque et al., 2020), rather than a binary judgment. Integrating such a score in our approach requires significant changes, but would offer the opportunity for a more nuanced moral-ladenness detection and preservation. We will acknowledge this as future work in the updated version of the paper.
>
> - **It would also be good to know how well LLM judges are able to approximate human judgements on this task, to facilitate automatic evaluation of moral framing preservation.**
>
> This would indeed be an interesting exploration; however, since our task and evaluation are novel, **we would nevertheless require extensive human evaluation** to validate the results of the LLM judge, as recommended in a recent large-scale empirical study [3]. Given the results in our paper, experimenting with the LLM-as-a-judge paradigm is a compelling direction for future work. If validated to be reliable, **such an approach could be integrated into our methods**, employing the LLM judge as a critic to iteratively improve the generated summary, à la RLAIF. We will acknowledge this as future work in the updated paper version.
>
> - **It would be good to provide some data on inter-annotator agreement.**
>
> Due to the structure and goal of our evaluation task, reporting annotator agreement is unfeasible. We would kindly redirect you to the comment to Reviewer RS8W, where we provide an extensive justification.
>
> - **Explain what is measured by each of the metrics in Table 2.**
>
> Section 4.4.1 describes the metrics used in Table 2. However, we recognize that an explicit reminder in Section 5.1 would facilitate the reader's interpretation of Table 2. We will add it in the updated version of the paper.
>
> [1] Kiesel et al., 2023, "SemEval-2023 Task 4: ValueEval: Identification of Human Values Behind Arguments." In *Proceedings of the 17th International Workshop on Semantic Evaluation (SemEval-2023)*, pages 2287–2303, Toronto, Canada. Association for Computational Linguistics.
>
> [2] Kiesel et al., 2024, "Overview of Touché 2024: Argumentation Systems". In *15th International Conference of the CLEF Association (CLEF 2024)*, volume 14959 of Lecture Notes in Computer Science, pages 308–332. Springer.
>
> [3] Bavaresco et al. "LLMs instead of human judges? A large scale empirical study across 20 NLP evaluation tasks." arXiv preprint arXiv:2406.18403 (2024).

---

> > ### Comment · Reviewer_iyXe · 2025-06-05
> >
> > Thank you for this informative response. I will maintain my (already high) score.

---

### Official Review · Reviewer_RS8W · 2025-05-10

**Rating:** 6
**Confidence:** 3
**Ethics Flag:** 1

**Summary:**

This paper presents an exploratory study on the preservation of moral framing in automatic summarization, based on the assumption that morally laden language plays a key role in how content is interpreted. The authors propose several zero-shot prompting strategies aimed at retaining morally laden words in summaries generated by three selected LLMs: LLaMA-3-70B, Command R+, and DeepSeek R1. Some settings involve explicitly providing the model with these words, either through automatic or manual annotation, and incorporating them into the prompt. The paper includes an extensive evaluation setup, combining automated metrics, crowdsourced judgments, and expert assessments, which yield a range of insights into the effectiveness of the proposed strategies.

**Questions To Authors:**

- The paper justifies the choice of LLaMA-3-70B, Command R+, and DeepSeek R1 based on their strong summarization capabilities. However, since this is an exploratory study, did you consider testing a broader range of models, including smaller ones? Doing so might yield interesting comparative insights into how model size and architecture affect moral frame preservation.
- The authors note that the Plain and Direct prompting strategies lead to shorter summaries. Did you consider controlling for summary length to ensure fairer comparisons across strategies?
- The results appear to be similar for both crowdworkers and experts: all prompting strategies perform comparably, except for Plain. Is there a justification for this result? It seems that the paper hypothesizes differences between the strategies, hence the motivation to explore them, so it would be helpful to better understand why the observed results did not diverge more clearly across prompting strategies.
- Did you experiment with any few-shot prompting strategies for this task? If not, could you comment on whether you see potential benefits or limitations in applying few-shot approaches to moral frame preservation?

**Reasons To Accept:**

- The paper addresses an interesting and underexplored aspect of summarization, focusing on the preservation of moral framing, which is a valuable contribution to the field.
- The authors propose several prompting strategies and conduct both human and automatic annotations, including expert evaluations that reveal valuable insights.

**Reasons To Reject:**

- While the authors state that their focus is on moral frame preservation, it would be valuable to consider how this quality dimension interacts with others, such as conciseness and comprehensiveness. There may be trade-offs involved, where the model needs to balance retaining moral framing with covering key content efficiently. Although the authors briefly hint at this in the results, a more explicit analysis of such trade-offs would strengthen the paper.
- Some details are missing that would help readers better understand and assess the work. For example, the inter-annotator agreement among the crowdworkers during evaluating summaries and highlighting spans is not clearly reported (as far as I understand, each article is annotated by two workers).

---

> ### Author Response · Authors · 2025-06-01
> **Rebuttal**
>
> We thank the reviewer for the valuable feedback.
>
> - **There may be trade-offs involved where the model needs to balance retaining moral framing with covering key content efficiently.**
>
> Indeed, our intuition, too, was that an improved moral framing preservation could come at the cost of other summary aspects such as conciseness. To control this, we prompt the models to limit the summary length to 200 words across all methods (line 99). Yet, **the resulting summaries vary in length across the methods and models**, with *Oracle* and *Class* typically leading to longer summaries, **as we discuss in Appendix D.1** (lines 637-647). However, we observe negligible or very weak correlation between the human evaluation results and the summaries' lengths (Table 5). We will gladly move this discussion to the main paper when additional space is available.
>
> - **The inter-annotator agreement among the crowdworkers is not clearly reported.**
>
> We would like to emphasize that **it is not feasible to measure inter-annotator agreement in our setting since annotators rated different items**. To elaborate, for each article, an annotator was asked to highlight morally framed spans of text in the article and then rate each highlighted span on a Likert scale. In this setting, the spans identified by two annotators for the same article are quite different (both in the number of spans and in the number of highlighted words within a span). Importantly, **making the two annotators agree on the spans of text to highlight is not the goal of our evaluation**; instead, the highlighting functions as a mnemonic mechanism to remember the morally framed bits of the article. Our aim is to judge the extent to which annotators consider the highlighted moral frames to be preserved in the summaries, which is reflected in the Likert scores that we report. We will stress this aspect in the updated version of the paper. We further elaborate on potential ways to measure consistency in annotators' ratings in the comment below.
>
> - **Did you consider testing a broader range of models, including smaller ones?**
>
> We examine whether it is even possible to prompt LLMs to preserve moral framing; since **there was no prior work on this aspect, we opted for large state-of-the-art models**. Now that we have evidence that this is feasible and of the involved trade-offs, **investigating the efficacy of smaller models is an interesting future direction**. Evaluating whether specific approaches (e.g., CoT or few-shot learning) can allow a smaller model to perform on par with or come close to the larger ones would provide valuable insight.
>
> - **Did you consider controlling for summary length?**
>
> As we elaborate in the answer above, we limited the summaries' length to 200 words (line 99).
>
> - **The results appear to be similar for both crowdworkers and experts. It seems that the paper hypothesizes differences between the strategies.**
>
> **The crowd and expert evaluations provide different types of insight.**
> Indeed, as discussed in lines 101-114, we anticipated differences between the strategies. Besides evaluating the difference between basic and word-preserving prompts, we asked: (1) Is a zero-shot approach sufficient (*CoT*), or are human labels necessary (*Class/Oracle*)? (2) Is supervised classification a good proxy for human labels (*Class*), or are direct human labels necessary (*Oracle*)? The crowd evaluation shows that *Class* and *Oracle* perform comparably and significantly better than the other methods, whereas *CoT* is indistinguishable from the baselines, thus answering *No* to the first question above and *Yes* to the second. This gives us valuable **insight into the amount of resources required** to achieve the goal of moral frame preservation.
> In contrast, the expert evaluation did not show significant differences between the four moral framing-sensitive strategies, likely due to the small sample. However, the justification analysis (Table 8, lines 287-303) shows **qualitative differences across the five methods**---e.g., that *CoT*, despite being ranked comparably to the other word-preserving approaches, commits moral framing addition more frequently.
>
> - **Did you experiment with any few-shot prompting strategies for this task?**
>
> We did not experiment with few-shot learning, but we agree that it is a **natural extension of the CoT approach** to improve the prediction of moral-laden words, as we suggest as future work (lines 328-330). Few-shot learning could also be used by **providing examples of "regular" and moral frame preserving summaries** in the prompt, guiding the model in recognizing the difference before generating a novel summary. This could be an interesting follow-up experiment using the summaries generated in this paper. Nevertheless, **a (possibly expert) validation** of the summaries used as examples is necessary to ensure human oversight.

---

> > ### Author Response · Authors · 2025-06-01
> > **Alternative way to measure annotation consistency**
> >
> > In this comment, we further elaborate on **an approach to measure the consistency of the annotators' ratings in the crowd task**. We apologize for using this extra comment, but we believe that an example of the annotation task is needed to clarify our explanation, thus requiring extra space. In any case, the main answer to the review is in the first comment.
> >
> > The table below shows a fictitious example of the crowd annotations for the summaries generated through one of the five methods.
> >
> > | Article ID | Annotator ID | Likert scores | Mean annotator score |
> > |:---|:---|:---|:---|
> > | 0 | a | [2,2,3,5] | 3 |
> > | 0 | b | [3,3,4] | 3.33 |
> > | 1 | c | [3,4,5] | 4 |
> > | 1 | d | [1,3,5,4,3] | 3.2 |
> > | ... | ... | ... | ... |
> > | 59 | x | [2,5] | 3.5 |
> > | 59 | y | [3,3,3,1,2] | 2.4 |
> >
> > In this example, two annotators (a and b) annotated article 0. Each of the two annotators highlighted a different number of morally framed spans in the article (4 and 3, respectively) and thus assigned different numbers of Likert scores. Calculating the inter-annotator agreement of these scores is not possible, as these scores were assigned to different items (i.e., different text spans). We then averaged these Likert scores, resulting in one score per annotator (3 and 3.33, respectively). Then, as reported in the table below, we average the two annotators' scores into one article score (3.17). We performed this for all 60 articles in the test set, and reported in Table 4 the mean and standard deviation of these 60 scores (in this fictitious case, we would report the mean and standard deviation of the 60-dimensional vector [3.17, 3.6, ..., 2.95]).
> >
> > | Article ID | Mean article score | STD article score |
> > |---|---|---|
> > | 0 | 3.17 | 0.17 |
> > | 1 | 3.6 | 0.4 |
> > | ... | ... | ... |
> > | 59 | 2.95 | 0.55 |
> >
> > Additionally, we can calculate the standard deviation between the two mean annotator scores assigned to one article (in the case of article 0 in this example, the standard deviation of the vector [3, 3.33], which is 0.17). We can then compute the mean and standard deviation of the standard deviation vector (in the fictitious example above, of the vector [0.17, 0.4,...,0.55]).
> >
> > The table below reports the mean and standard deviation of the standard deviation vector of our experimental results across the five methods.
> >
> > | Method | Mean STD |
> > |---|---|
> > | *Plain*   |  0.80 $\pm$ 0.53 |
> > *Direct*  |   0.59 $\pm$ 0.47 |
> > *CoT*      |   0.59 $\pm$ 0.46 |
> > *Oracle*  |   0.55 $\pm$ 0.45 |
> > *Class*   |  0.57 $\pm$ 0.48 |
> >
> > These results show that the *Plain* approach leads to larger differences in average Likert scores among annotators when compared to the other approaches, which are comparable to each other. Furthermore, the magnitude of the mean standard deviation appears relatively low (although difficult to judge as no baseline is available), hinting at a consistency among annotators.
> > Since, as motivated in the comment above, we believe that this is not central to our evaluation, we refrained from introducing it in the paper to avoid diverting the reader's attention. However, we can add this explanation and the table above to the Appendix.

---

### Official Review · Reviewer_thoM · 2025-05-13

**Rating:** 8
**Confidence:** 5
**Ethics Flag:** 1

**Summary:**

This paper tackles the task of summarizing news articles by maintaining framing, which is the act of writing in morally charged language rather than using neutral terms. Specifically, the authors study the preservation of moral framing in AI-generated news summaries.

They employ straightforward approaches, each extracting the morally loaded words and then summarizing the article. More precisely, they experiment with five diffferent prompts: 1) plain (summarize only), 2) direct (1 prompt to summarize and preserve the moral framing instruction ), 3) CoT (first identify moral-laden words, then summarize), 4) Class (identify moral laden words using a trained classifier, then summarize) and 5) Oracle (the moral-laden words are provided as part of the instructions to summarize.)

Their experiments are thoroughly conducted using three big open-weight models. For each, they repeat an article summary several times and report their results.

The authors conduct an extensive analysis including automatic crowdsourcing and expert evaluations, revealing interesting results.


In general, the work is well motivated, the approach is reasonable, and the analysis is extensive, yielding several important findings. However, the paper has room for improvement in the camera-ready version by highlighting all the findings (bold, restating more findings in the intro or conclusion).

**Questions To Authors:**

Notes to authors, in addition to the points mentioned above:

- Mention some of the ten moral foundation theory elements (line 124), and give an additional dataset example.
- Table 2 caption: explicitly mention the meaning of each metric and avoid abbreviations when there is enough space.
- Line 125 models could be better organized. For example, the details of the classifier could be moved to a "paragraph." Also, mentioning a metric value for the two approaches used for training the classifier would be helpful.
- The following are not clear:
  > "line 148 -  We randomly choose a seed whose results are used for human evaluation"

  > "line 150 -We do not distinguish among the ten MFT elements in the summary generation, but we do in the evaluation."

- Map findings to the table more explicitly. For example, line 248 (Table 4, Likert score)

**Reasons To Accept:**

- The paper tackles a very important task: to empower AI-generated summaries to preserve *framing*, which implicitly includes the writer's implicit intentions
- The approach is straightforward, simple, and thorough
- The analysis is extensive and reveals very interesting (and in some cases, unexpected insights

**Reasons To Reject:**

(can be solved in camera ready version)

-  The paper has room for improvement by improving the structure of the results section:
     > as simple improvement such as highlighting all the findings (bold, restating more findings in the intro or conclusion), can help improve the readability.

---

> ### Author Response · Authors · 2025-06-01
> **Rebuttal**
>
> We thank the reviewer for concrete recommendations on how to improve the paper.
>
> Below, we clarify the two unclear sentences.
>
> > line 148 - We randomly choose a seed whose results are used for human evaluation.
>
> We repeated the generation of each summary five times with five different seeds and reported the average as automated evaluation results. However, in the human evaluation, it was feasible to evaluate only one of the five summaries generated with the five seeds; thus, we randomly chose one of the five seeds and evaluated the summaries generated with that seed.
>
> > line 150 - We do not distinguish among the ten MFT elements in the summary generation, but we do in the evaluation.
>
> The EMONA dataset contains token-level annotations of the ten MFT elements (i.e., a token can be annotated as non-moral or with one of the ten MFT elements). In our methods, we aggregated the ten MFT labels into one "moral-laden" label and thus treat each token as being either moral-laden or not. The MD (Moral Divergence) metric measures the divergence between the distribution of moral annotations in the article and the summary, considering all 10 labels. This aims to measure whether the model preserves only words annotated with one specific MFT label or preserves words annotated with all types of MFT labels. We will add a forward reference to Section 4.4.1 to clarify our statement.

---

> > ### Comment · Reviewer_thoM · 2025-06-03
> > **Acknowledging Response**
> >
> > Thank you for your response. I will maintain my score.

---

### Official Review · Reviewer_zVNp · 2025-05-14

**Rating:** 5
**Confidence:** 4
**Ethics Flag:** 1

**Summary:**

This paper presents the first study on the preservation of moral framing in AI-generated news summaries. The authors argue that moral-laden language, intentionally used or quoted by journalists, is a key aspect of news framing that should be recognized and retained by summarization models. To address this, they propose prompting strategies that first identify morally charged words—via chain-of-thought reasoning or supervised classification—and then guide large language models to preserve such words in the generated summaries. The study evaluates three LLMs and five prompting methods on the EMONA dataset, using automated, crowd-sourced, and expert evaluations. Results show that the proposed approach improves moral framing preservation without compromising overall summary quality.

**Questions To Authors:**

See reasons to reject.

**Reasons To Accept:**

1. The motivation of this paper seems clear and sound.

2. The empirical results also give evidence of the effectiveness of taking moral framing into account.

**Reasons To Reject:**

1. A key concern is the lack of quality control in the human annotation process. Given the subjectivity of moral framing, the absence of validation measures (e.g., annotator qualification or agreement checks) raises doubts about the reliability of the foundation of this paper's evaluation.

2. The experiments could be more comprehensive. For example, it would be valuable to explore whether few-shot learning significantly improves the accuracy of moral-laden word prediction and subsequently enhances summarization quality. Additionally, the evaluation dataset is relatively small, which may limit the generalizability of the findings.

---

> ### Author Response · Authors · 2025-06-01
> **Rebuttal**
>
> We thank the reviewer for the interesting comments.
>
> - **The absence of validation measures raises doubts about the reliability of this paper's evaluation.**
>
> We would like to clarify that **we introduced qualification requirements as well as attention checks** (as detailed in Appendix C.1, lines 599-608). We only considered crowd workers who had completed at least 100 Prolific jobs with at least a 95\% acceptance rate, and passed four attention checks spread over the annotation task. Furthermore, we conducted an additional evaluation with expert annotators (whose credentials are reported in Appendix C.2), whose qualified qualitative input provided insights that had not emerged from the crowd evaluation (as discussed in Section 5.3).
>
> Next, we would like to emphasize that **it is not feasible to measure inter-annotator agreement in our setting since annotators rated different items**. To elaborate, for each article, an annotator was asked to highlight morally framed spans of text in the article and then rate each highlighted span on a Likert scale. In this setting, the spans identified by two annotators for the same article are quite different (both in the number of spans and in the number of highlighted words in a span). Importantly, **making the two annotators agree on the spans of text to highlight was not the goal of our evaluation**; instead, the highlighting functioned as a mnemonic mechanism to remember the morally framed parts of the article. What we sought was to judge the extent to which annotators considered the highlighted moral frames to be preserved in the summaries, which is reflected in the Likert scores that we report. We will stress this aspect in the updated version of the paper. We further elaborate on potential alternatives to measure consistency in annotators' ratings in the response to reviewer RS8W.
>
>
> - **It would be valuable to explore whether few-shot learning significantly improves the accuracy of moral-laden word prediction.**
>
> We recognize that our approaches can be enhanced to improve moral frame preservation. However, we would like to emphasize that **our contribution is more fundamental in that we examine whether it is even possible to prompt LLMs to preserve moral framing**. To this end, we compared five intuitive methods and **focused on evaluating the results**. Given our promising results showing that moral frame preservation in LLM-generated summaries can be achieved, additional approaches can be explored to address the demonstrated trade-offs (e.g., the need for training a supervised classifier). Few-shot learning is indeed a natural extension of the *CoT* method to improve the prediction of moral-laden words, as we suggest as a future direction (lines 328-330).
>
> - **The evaluation dataset is relatively small, which may limit the generalizability of the findings.**
>
> We agree that a larger evaluation dataset would lead to more generalizable findings. However, we would like to stress that (1) we performed a **stratified sampling** of the dataset to ensure that a diverse sample of articles would be represented in the human evaluation dataset (lines 138-141). Next, (2) we decided to **focus our resources on the depth of the human evaluation**, conducting a two-fold human evaluation to ensure that the findings are consistent across different types of subjects (laypeople and experts). Finally, (3) we conducted a thorough **statistical analysis** to ensure that our findings are sound despite the relatively small evaluation dataset size (as reported in Tables 4 and 7).

---

> > ### Comment · Reviewer_zVNp · 2025-06-10
> > **Acknowledging Response**
> >
> > I will maintain my score.

---

### Decision · Program_Chairs · 2025-07-08

**Decision:**

Accept

**Comment:**

The authors present a study regarding the preservation of moral framing in automatic news summary generation using LLMs. Specifically, the propose a sequential prompting mechanism of: 1) identifying morally charged words (either via explicit classification or chain-of-thought reasoning) and 2) guiding LLMs to prefer these words during summary generation. From an empirical perspective, they evaluate three LLMs with five prompting methods (variants of above and simpler baselines) on the EMONA dataset with automated, crowd-sourced, and expert evaluation to demonstrate preservation of moral framing without reducing overall summary quality.

Consensus strengths of this work identified by the reviewers include:
- The problem of moral framing hasn't been studied sufficiently in the context of automatic summarization, which is critical to faithful news summaries. It is likely of significant academic and likely commercial interest.
- The proposed approaches are enabled by SotA LLMs for which the authors develop straightforward, but intuitively appealing and effective methods.
- The empirical results demonstrate the ability of LLMs to preserve moral framing and the analysis/interpretation is interesting. Human annotations add confidence and lead to interesting findings.
- The paper is well-motivated and clearly written.

Consensus limitations of this work identified by the reviewers include:
- A lack of rigor in the human annotation evaluation in terms of inter-annotator agreement. While the authors are correct that this analysis isn't trivial for structured tasks (as elaborated during rebuttal), the tasks can be decomposed, latent variable analysis can be used with sufficient annotations, etc. With the current analysis, I am convinced that the findings are valid and prefer to have these results (on a more interesting task) disseminated over waiting for perfection.
- The reviewers recommended experiments around few-shot learning, lexicon-based methods, etc.
- More granularity in the evaluation of summary quality such that the tradeoffs regarding preserving moral framing can be more granular also.

Overall, I believe the consensus amongst the reviewers is that the clarity and significance of this work clearly warrants publication, while the quality is sufficient and the originality is notable. The primary concerns are around inter-annotator agreement analysis, which has been partially addressed in rebuttal and I don't believe should be a blocker to publication.